# Examining the Association between Sports Participation and Mental Health of Adolescents

**DOI:** 10.3390/ijerph192417078

**Published:** 2022-12-19

**Authors:** Asaduzzaman Khan, Kazi R. Ahmed, Tarissa Hidajat, Tracy Kolbe-Alexander, Elizabeth J. Edwards

**Affiliations:** 1School of Health and Rehabilitation Sciences, The University of Queensland, Brisbane 4072, Australia; k.ahmed@uqconnect.edu.au; 2School of Education, The University of Queensland, Brisbane 4072, Australia; t.hidajat@uq.net.au (T.H.); elizabeth.edwards@uq.edu.au (E.J.E.); 3School of Health and Medical Sciences, Centre for Health Research, University of Southern Queensland, Ipswich 4305, Australia; tracy.kolbe-alexander@unisq.edu.au; 4UCT Research Centre for Health Through Physical Activity, Lifestyle and Sport (HPALS), Division of Research Unit for Exercise Science and Sports Medicine, University of Cape Town, Rondebosch 7700, South Africa; 5Manna Institute, Centre for Health Research, University of Southern Queensland, Springfield Central 4300, Australia

**Keywords:** adolescents, sports, depression, life satisfaction, Bangladesh

## Abstract

Sports participation has been linked to various health outcomes; however, there is scant literature exploring this relationship in developing countries. We used a mixed method approach to examine the association between sports participation and mental health of Bangladeshi adolescents (*n* = 320; 13–17 years; 59% boys) and to explore sports preferences and barriers to sports participation across genders. A survey collected team and non-team sports participation, depression, and life satisfaction. Focus groups (16 boys, 16 girls) explored preferences for, and barriers to, sports participation. Regression analysis showed that higher team and non-team sports participation were associated with lower depressive symptoms in boys (β = −1.22, 95% CI:−2.55 to −0.10; β = −2.50, 95% CI:−3.83 to −1.16, respectively), while greater participation in team sports was associated with less depression in girls (β = −2.44, 95% CI:−4.63 to −0.24). Participation in team and non-team sports was positively associated with life satisfaction for boys and girls. Boys reported preferences for playing football and cricket, while girls favored skipping and running. Prolonged time on electronic devices was reported as barriers to sports participation in both genders. Furthermore, household chores, family restrictions, and unsafe environment were reported by girls. Participation in team sports may provide mental health benefits for both genders, while non-team sports may be more beneficial to boys than girls.

## 1. Introduction

Sport is defined as a subset of physical exercises undertaken individually or as a part of a team [1]. Sports participation has many well-known health benefits in children and adolescents [2]. Participation in sports, including team or individual sports, has been linked with better physical and mental health and personal development, as well as reduced weight and greater physical activity [1,3,4]. The health benefits of sports participation are widespread, ranging from enhancing emotional health to promoting social relationships, fostering resilience and self-esteem, and decreasing risks of chronic diseases and disability [1,5]. In sum, active sports participation has been shown to bring better outcomes across the lifespans of adolescents regarding physical, mental, social, and emotional health [6]. Specifically, a recent review noted that adolescents who participated in team sports appeared to have better psychosocial health, such as lower anxiety, depression, and social problems, compared to those who participated in non-team sports [7]. The review concluded that participation in team sports during adolescence was associated with fewer symptoms of depression and anxiety at a later stage. Moreover, organized sport, which involves rules and formal practice, competitive in nature, played in a team or as an individual, has been reported as one of the most popular forms of leisure-time physical activity worldwide [8]. In accord, a meta-analysis reported that adolescents who played organized sports had lower symptoms of depression and anxiety compared to those who did not participate in sports [9].

Despite the well-established benefits of sport participation in health and wellbeing, the quality of participation in sports varies across countries. For example, participation in sports among adolescents is often higher in Western high-income countries, with annual participation rates between 60% and 80%, and rates varied substantially by age, sex, and ethnicity [8]. However, the sparsely available data on sport participation in many low- and middle-income countries indicate a relatively lower level of participation [10]. For example, only 17% of Chinese children and adolescents aged 9–17 years participated in organized sports during the past 12 months [11], while self-reported organized sports participation was 37.3% among children and adolescents in Thailand [12].

Adolescent mental health problems are an ongoing public health concern, with approximately 10–20% global prevalence [13]. Almost 50% of all adolescent mental health and behavioral problems start by the age of 14 years, with depression and anxiety being the most prevalent [14]. Examining data from 59 low-income and middle-income countries, one study [15] found that the overall prevalence of suicide ideation was 17%, suicide planning was 17%, and suicidal ideation was also 17% in the 12 months prior to the survey; with girls having a higher prevalence than boys for suicidal ideation (18.5% vs. 15.1%), planning (18.2% vs. 15.6%), and attempts (17.4% vs. 16.3%). In terms of non-Western adolescents, one in every seven young people in Sub-Saharan Africa reported to have major psychological problems [16], and in China, the prevalence of one or more psychiatric disorders was approximately 19% in boys and 16% in girls [17]. One study from Bangladesh showed mental ill-health in children and adolescents aged 2–16 years ranged from 13.4% to 22.9% [18], while another study in urban settings reported that 25% of adolescents aged 13–16 years had depressive symptoms, with girls having more symptoms than boys [19]. Although mental disorders are widespread among children and adolescents in many developing countries, it was estimated that over 90% of global mental health resources are being spent in developed countries that only represent 5% of the global population [20]. 

Most research in sports participation and potential links with health and wellbeing are from Western or developed countries, with little representation of developing countries [21]. There is limited research in adolescent mental health in Bangladesh [19,22] and to date no studies have examined sports participation or whether participation in sports is associated with any health and wellbeing parameters of Bangladeshi young people. The aims of the present study were: (i) to examine the relationship between sports participation and mental health of Bangladeshi adolescents, and (ii) to explore sports preferences and barriers to sports participation of both boys and girls.

## 2. Materials and Methods

We used a mixed method approach. Ethical approval was obtained from the Human Research Ethics Committee of The University of Queensland, Australia (2018000885, 31 May 2019), and the Bangladesh University of Health Sciences (BUHS/BIO/EA/18/12, 26 December 2018). All participants provided written parental consent and written student assent in simple Bengali language (local language) and data were deidentified.

### 2.1. Quantitative Examination

#### 2.1.1. Participants 

Baseline data from a large trial, conducted in March 2019 in urban school settings in Dhaka city, Bangladesh, were used to examine the associations [23]. A total of eight secondary (or high) schools took part in the trial, with a minimum of 40 students from Grade 8 and 9 (aged 13–17 years) from each school. For a school with more than 40 students in Grade 8 and 9, a random selection was used to recruit the required sample size (*n* = 40 students per school). Thus, a total of 320 students completed a self-administered survey and are included in the current analysis. 

#### 2.1.2. Outcome Measures

Adolescents’ mental health was assessed with one negative (e.g., depressive symptoms) and one positive (e.g., life satisfaction) assessment. Depressive symptoms were measured using the 10-item Centre for Epidemiologic Studies Depression Scale (CESD-10), as used previously in Bangladeshi adolescents [19]. The CESD-10 items include: (1) I was bothered by things that usually don’t bother me; (2) I had trouble keeping my mind on what I was doing; (3) I felt depressed; (4) I felt that everything I did was an effort; (5) I felt hopeful about the future; (6) I felt fearful; (7) My sleep was restless; (8) I was happy; (9) I felt lonely; and (10) I could not “get going”. Response options range from 0 to 3 for each item: 0 = rarely or none of the time, 1 = some or little of the time, 2 = moderately or much of the time, 3 = most or almost all the time). A total score was obtained by summing the 10 items and ranged from 0 to 30, with a higher score suggesting a greater severity of symptoms. In the present study, the internal consistency of the CESD-10, measured by Cronbach’s alpha, was 0.84. Life satisfaction was assessed using the single-item Cantril ladder, which has demonstrated reliability and convergent validity in adolescents [24]. Participants responded on a visual analog scale, ranging from the worst possible life (0 points) to the best possible life (10 points).

#### 2.1.3. Study Factors

Participants were asked about their involvement in sports participation with two items: “Are you involved with any team sports (e.g., football, volleyball, and cricket) in your school?” and “Are you involved with any non-team sports (e.g., athletics) in your school?” with two response options: Yes and No. 

#### 2.1.4. Covariates

A set of covariates was collected including age, sex, school, and family-level data (e.g., parental education, family income data). BMI z-scores were computed using objectively collected weight and height data reported by the participants. Time spent on social media per day was also captured. 

#### 2.1.5. Data Analysis

Proportions of adolescents involved with team and non-team sports at schools were computed. Given the demonstrated sex differences in both outcomes and study factors, the analyses were stratified by sex. As the outcome variables were continuous scores, multiple linear regression modeling was used to examine the associations of team and non-team sports participation with depressive symptom and life satisfaction scores, adjusted for age, mother’s education, family income, BMI z-scores, and duration of social media use per day. Father’s education was dropped from the modeling due to its high correlation with mother’s education. The association estimates are presented in the form of regression coefficients and their 95% confidence interval (CI). All analyses are conducted using Stata 17SE [25]. 

### 2.2. Qualitative Exploration

#### 2.2.1. Participants 

Four focus group discussions (two for boys and two for girls) were conducted with students of grades 8–9 (aged 13 to 17 years) between January–February 2019. Recruitment flyers were distributed to students across four schools. Students were asked to express their interest in participation to their class teachers. In the event of more than eight interested students in one school, eight participants were randomly selected from the pool of interested students. Finally, sixteen girls and sixteen boys (*n* = 32) were provided with participant information sheets and provided parental consent forms and student assent for participation. All focus groups were conducted during school hours, in a separate school room with the appropriate permission of the respective school authority without the presence of any schoolteacher or staff.

#### 2.2.2. Procedure

A guide was developed for the focus groups based on the relevant literature review. The guidelines underwent field testing among a small group of male and female students from a secondary school who were not enrolled in the study. The focus group discussions encouraged students to express their understanding of sports participation and provide details of their preference for sports, and perceived barriers to participating in sports. Discussions were conducted in the Bangla language. Students uttered their participant identification number each time, before expressing their opinions. Focus group discussions lasted approximately 30 min, were recorded, and notes were taken with the permission of the participants. 

#### 2.2.3. Data Management and Analysis

Focus group recordings were transcribed in Bangla and the scripts were translated into English by a bilingual expert. The English version of the scripts was translated back to Bangla by another bilingual expert in order to retain the real meaning of the original statements and to ensure validity of translation. A content analysis was conducted to determine the underlying themes. The research team independently reviewed the data, made a list of keywords and contents that emerged through the data, and created broader categories based on concepts of similar meaning (e.g., content categories). The content categories were listed in order of frequency of occurrence, with more frequently occurring categories indicating a higher level of significance. Themes and subthemes were identified after a detailed examination of the narrative texts. Common themes were identified, and keywords and phrases from the text were used to mark the themes throughout the text. Themes were identified under two main sections: sports preferences and perceived barriers to sports participation.

## 3. Results

### 3.1. Quantitative Findings

A total of 320 students completed the survey (*M_age_* = 14.30 years, *SD_age_* 1.04 years) and 59% were boys. Approximately 11% of participants reported involvement with team sports, with boys doing more than girls (12.2% vs. 7.6%), while 10% reported involvement with non-team sports in schools, with no gender differences (boys: 10.6% vs. girls: 9.1%). Results of multiple linear regression modeling are presented in Table 1. The modeling showed that both team and non-team sports participation was inversely associated with depressive symptoms in boys (adjusted β = −1.22, 95% CI: −2.55 to −0.10, and adjusted β = −2.50, 95% CI: −3.83 to −1.16, respectively), while participation in team sports was only inversely associated with depressive symptoms in girls (adjusted β = −2.44, 95% CI: −4.63 to −0.24), after adjusting for a set of covariates (see Table 1). The modeling also showed that both team and non-team sports participation were positively associated with life satisfaction in both boys and girls, with boys having slightly stronger association estimates than girls. For example, adjusted association between team sports participation and life satisfaction was β = 1.72 (95% CI: 1.04, 2.40) for boys, and β = 1.47, 95% CI: 0.38, 2.57) for girls.

### 3.2. Qualitative Findings

Thirty-two students participated in the focus group discussions (*M_age_* = 14.00 years, *SD_age_* = 0.76 years) and 50% were boys. After exploring the qualitative data, three main themes were identified: sports preferences, perceived barriers, and access to opportunities. Figure 1 presents a thematic map of the themes and sub-themes.

#### 3.2.1. Theme 1: Sports Preferences 

Content analyses of the focus group discussions showed some gender differences for preference for sports participation. In general, most of the participants preferred outdoor sports, such as cricket, football, badminton, volleyball, and hide and seek. Some also preferred indoor sports, such as board games, traditionally known as Ludo, Carrom, and Chess. Girls preferred sports such as skipping games and traditional sports (e.g., sewing sports/candle sports/phultoka/gollachut/truth or dare/kanamachi/musical chair) which involve running at a slow pace and are physically less demanding. Girls reported spending their recess time at schools talking to their peers rather than participating in sports activities. Some girls also preferred volleyball, handball, and badminton during the winter season and others preferred board games (e.g., Ludo) during their leisure time. Boys mostly preferred sports such as football and cricket, followed by badminton and volleyball. Boys usually preferred competitive sports which are more physically demanding, and their preferred sports did not change throughout the year. Participants were also asked whether they would like to be involved in same-sex or mixed groups while engaging in any sports. Girls noted that they would rather participate in sports with other girls, whereas all the boys were open to participating in mixed-gender group sports.

#### 3.2.2. Theme 2: Perceived Barriers

Common perceived barriers to sports participation across genders included extended time on digital screens, hectic daily schedules, extra academic pressure, not having enough space for playing or jogging, and laziness. Spending time on electronic devices (e.g., television, mobile phone, computer, playing videogames) was highlighted by most of the participants as the main barrier to sports participation. Four subthemes were identified:

Subtheme 1: Behavioral factors. Participants stated that behavioral factors, such as laziness, timidity, and fear of accidents, were barriers for sport participation. Few participants reported spending leisure time on computers with their siblings at home. Boys mentioned a preference for watching television and video games, while girls preferred using mobile phones over sports. Girls also reported spending a lot of time on social media and screen-based activities, which reduced time for play. Girls also felt that they had fewer sports options available at home that encouraged them to use digital screens.

“Our friends are passing their time at home involving with new projectors, computer games, mobile game, Facebook. Therefore, they could not manage time for playing sports.” (Girls, Focus Group 2)

“Our laziness may be the reasons for not playing sports. There is assembly in schools but many of my friends are reluctant to join the assembly despite reminding them. We simply can’t accuse schools for not creating opportunities ….”(Girls, Focus Group 1)

Subtheme 2: Overloaded curriculum. Participants reported high academic pressure as one of the main barriers to sports participation. They stated after-school activities usually consisted of homework followed by unstructured behaviors, such as television viewing or using a mobile phone. In addition to attending academic activities at school, students described needing to attend after-school private tutoring and other extra-curricular (non-sports) activities, such as playing music and singing. As such, students reported no time to engage in indoor or outdoor sports activities. High academic pressure was also reported to increase their stress, which was sometimes managed by using social media. 

“The biggest reason for not participating in any sports is that we do not get sufficient time after school. Because of coaching and tuition at home, helping family for chores, we do not get time to play.”(Girls, Focus Group 1)

“We go for private tuition/coaching after school. Then we read ourselves at home.”(Boys, Focus Group 2)

Subtheme 3: Cultural norms. Sociocultural barriers to not engage in any sports are more prevalent among girls than boys. Girls reported that they have many restrictions on their movements and often are not allowed to go out for play. For example, lack of family support and domestic responsibilities (e.g., household chores) were reported as barriers to engage in sport. Girls also reported not having enough places to play indoors. Instead of going out for sports, a few participants described that they spent time on their smartphones talking or chatting with friends. Some girls stated that a culture of outdoor sports is not common in their society.

“Parents prohibit us. I mean they don’t allow us to go out and play…. Not even in the rooftop.”(Girls, Focus Group 1)

“We grew up, parents do not let us go out. There are many boys out there who can disturb, they do not want to allow us to play as we are girls.”(Girls, Focus Group 2)

“Our life has become a captive life as grow up ….. parents don’t allow us to go out in on order to protect us from being bullied by street boys and also to follow the tradition of the society.”(Girls, Focus Group 1)

Subtheme 4: Environmental/institutional factors. The lack of facilities at schools and after school seemed to affect the engagement of adolescents in sports participation. Some adolescents reported difficulties engaging in sports in school due to limited sport facilities (e.g., lack of playground, sports equipment), short recess time, sports day only once a year, and limited after-school activities. 

“Recess time is too short. So, we do not get any time to play.” (Girls, Focus Group 1)(Girls, Focus Group 1)

“We usually eat by talking and sitting in the classroom for our short recess time.”(Boys, Focus Group 2)

Participants reported that their once-a-week physical education class (e.g., 50 min/week) included theoretical health education components (e.g., health and hygiene knowledge) rather than active participation in sports. Additionally, classes are typically conducted in a classroom setting where students are sitting and memorizing facts and figures, which limits their opportunity to participate in sports. Students felt physical education classes were missed opportunities for sports participation. 

“Activity is done one day a week…Monday. We are given ideas of physical activities from the textbook in a physical education class. We are just taught from the textbook….. No extra care is given, or opportunity is created to encourage us to participate in sports at school, and there is no value attached with participation in sports.”(Boys, Focus Group 1)

Outside the school environment, safety concerns were cited by girls as barriers. Many girls mentioned that they were not allowed to go outside to play after-school as there are not dedicated sports facilities for girls. Girls reported there are limited sports facilities and mostly occupied by boys and male adults, which disadvantages girls’ participation in outdoor sports. 

#### 3.2.3. Theme 3: Access to Opportunities 

Many participants shared their views about opportunities to enhance sports participation. Students agreed unanimously that schools can play a vital role in promoting sports participation. Some of the main initiatives reported many participants included opening different sports clubs, creating and managing playing fields, dedicated sports facilities for girls (or dedicated time for girls), organizing periodical competitive sports programs and motivational counseling to students who are disinclined to sports, and advising parents to encourage their children to participate in sports. 

Both boys and girls agreed that physical education classes can create an important opportunity for sports participation in schools. Although, some schools offer only one class per week and other schools do not include any because physical education is not compulsory in the national curriculum in Bangladesh. Many participants felt that scheduling physical education classes at least twice times a week and ensuring the classes present additional opportunities for sports participation could promote sports in school settings.

“It would be better if physical education class would be held for two days instead of one day.”(Boys, Focus Group 2)

Some participants suggested that active parents could be role models and facilitate sports participation. They also shared that their parents could participate in sports with them, organize sports equipment at home, support them to play outside during their free time or weekend, all of which can encourage them to be more active.

## 4. Discussion

The present study is the first to investigate whether team or non-team sports participation was related to depressive symptoms and life satisfaction in Bangladeshi adolescents. Findings concur with previous studies that were primarily conducted in Western or developed countries [21], as well as several studies conducted in developing countries that found higher sports participation was associated with better mental health outcomes, such as lower depressive symptoms and higher life satisfaction [7,9,26,27]. More specifically, the current study found that both team and non-team sports were associated with lowered depressive symptoms in boys, while only team sports were associated with lowered depression in girls. Bangladeshi boys were recorded to have higher sports involvement than girls in the current sample, also aligning with results from other countries [28,29]. 

The influence of sports participation on depressive symptoms has been attributed to biological, psychological, and social mechanisms. In terms of biological mechanisms, the physical activities component in sports have been suggested to lead to lowered levels of stress-inducing hormones—such as adrenocorticotropin, cortisol, catecholamine, and epinephrine—while increasing levels of noradrenaline, serotonin, and endorphin hormones that have been associated with lowered levels of depressive symptoms [30,31,32]. In terms of social mechanisms, adolescents’ participation in team sports may provide increased opportunities for social interactions [9,33]. Past cross-sectional studies in adolescents also suggested that participating in team sports may increase a sense of social acceptance from receiving coaching and may heighten levels of social support from peers and adults [34,35]. Further, the psychological mechanisms underlying positive influences of sports participation on depressive symptoms have been attributed to elevated levels of self-esteem, self-confidence, and positive body image, from increased self-efficacy after accomplishing physical challenges and improved body awareness [33,36,37,38]. 

Interestingly, we found gender-based differences in the associations between team and non-team sports participation on depression, with participation in team sports linked to lowered levels of depressive symptoms in Bangladeshi adolescent girls. Our focus group discussions suggest that girls preferred to socialize with friends (e.g., through social media) rather than participating in sports, especially in outdoors, which is similar to adolescent girls in the United Kingdom [39]. One explanation for differences between girls–boys might be that girls have previously reported feeling incompetent with physical activities and often experience teasing from same- and opposite-sex peers while participating in sports, which has deterred them from sports participation [40]. Hence, it has been suggested that adolescent girls need supportive friends around them when participating in team sports to provide a sense of security, enjoyment, and encouragement [38]. One study suggested that team sports generate higher levels of social support from friends and family for girls, compared to participation in individual sports, which in turn may increase engagement with physical activity [41]. The team environment may influence adolescent girls’ self-esteem from the opportunities to engage with friends in accomplishing a collective goal [42], compared to individual sports [43]. Furthermore, participation in team sports in adolescent girls might be related to increased self-esteem due to heightened engagement and social support, which is also linked with lowered levels of depressive symptoms [41,42,43]. 

Our data showed that participation in team and non-team sports was associated with higher levels of life satisfaction in Bangladeshi adolescents. This finding is consistent with previous studies in Czech and Italian adolescents which found higher levels of life satisfaction linked to physical activities, such as team or non-team sports [44,45,46]. This association was suggested to have stemmed from the biological, psychological, and social benefits of physical activities on wellbeing [44,45]. One longitudinal study proposed that physical activity acts as a buffer between stress and life satisfaction in adolescents [47]. However, while higher life satisfaction has broadly been liked to greater participation in physical activities, including through sports [44,45,46], a dose-dependent relationship is yet to be determined [45]. Hence, the reciprocal nature of the relationship between life satisfaction and sports participation warrants further investigation. 

Our focus group discussions revealed gender differences in the barriers and preferences for sports participation among Bangladeshi adolescent boys and girls. Specifically, boys preferred competitive, more physically demanding sports, while girls preferred lower-intensity sports. Acknowledging the gender-based differences in sports preferences is crucial as they may promote sports involvement in adolescent boys and girls [28]. To promote sports participation, our students suggested a number of initiatives, including more physical education classes at school and classes including playing sports, creation of gender-friendly sports facilities and sports clubs, and organizing more sports competitions at schools. Families can also be instrumental in promoting sports participation through enabling a sporting environment at home by supplying sports equipment and supporting after-school sports activities. 

The present study has several limitations, including a small sample and self-reported data. Depressive symptoms were assessed using the self-reported CESD-10 scale, which has acceptable psychometrics but has not been validated among Bangladeshi adolescents. The cross-sectional design precludes determining causality of the relationships. Nonetheless, our finding that, one in ten Bangladeshi adolescents participate in sports with girls doing less than the boys, underscores the need for strategies to promote sports participation, especially among girls. Longitudinal studies are warranted to understand the inter-relationships of these variables, and their causal link with various health outcomes.

## 5. Conclusions

Overall, the current findings were consistent with previous studies in adolescents, where greater sports participation was related to better mental health outcomes, especially lower depression and higher life satisfaction. Our research implies the need to promote and facilitate sports participation as one of the means to maintain and improve adolescent mental wellbeing. Most importantly, our results contribute to the growing evidence of these relationships in developing countries and resource-poor settings. Our findings support the notion that both school and family can play roles in promoting sports participation of adolescents. Gender differences in preference for, and barriers to, sports participation can inform future research as well as strategies to promote sports participation in adolescents in Bangladesh and other countries that share similar culture and tradition. Future research including robust trials is required to understand the mechanisms of these findings.

## Figures and Tables

**Figure 1 ijerph-19-17078-f001:**
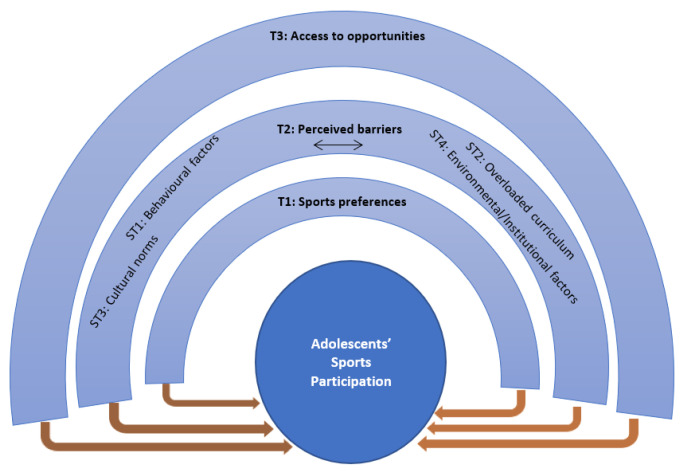
Thematic map of identified themes for sports participation (T: theme; ST: subtheme).

**Table 1 ijerph-19-17078-t001:** Adjusted associations of sports participation with mental health indicators in adolescents in Bangladesh.

Characteristics	Depressive Symptoms		Life Satisfaction	
	Boys	Girls	Boys	Girls
	β (95% CI)	β (95% CI)	β (95% CI)	β (95% CI)
Involved with team sports	−1.22 (−2.55, −0.10)	−2.44 (−4.63, −0.24)	1.72 (1.04, 2.40)	1.47 (0.38, 2.57)
Involved with non-team sports	−2.50 (−3.83, −1.16)	0.75 (−1.41, 2.91)	1.63 (0.92, 2.34)	1.09 (0.6, 2.12)

Notes: β adjusted regression coefficient; CI Confidence interval models were adjusted for age, body mass index, duration of social media use, mother’s education, family income.

## Data Availability

Data will be available on request.

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
