# Peer review of "Examining the Association between Sports Participation and Mental Health of Adolescents"

_ijerph, 2022, doi:10.3390/ijerph192417078_

Round 1

Reviewer 1 Report

Main comment: This is an interesting and well-written paper, supporting the commonly accepted idea that regular physical activities can improve quality of life, health and well-being, especially in children and adolescents. According to your results, sports participation is specifically associated with less depression symptoms and more life satisfaction in adolescents. You also explore importants issues about sports participation, related to gender and environnemental factors in the context of Bengladesh, using a mixed model design. 

Minor comments:

Minor spell check required line 345 and 380.

Depression scale and life satifaction scale may be added as additionnal material. If not possible, a short specific description of items included in each scale could be provided in order to improve methodology description.    

Author Response

Main comment: This is an interesting and well-written paper, supporting the commonly accepted idea that regular physical activities can improve quality of life, health and well-being, especially in children and adolescents. According to your results, sports participation is specifically associated with less depression symptoms and more life satisfaction in adolescents. You also explore importants issues about sports participation, related to gender and environnemental factors in the context of Bengladesh, using a mixed model design. 

Response: We thank the reviewer for their appreciative comments about our work.

Minor comments:

Minor spell check required line 345 and 380.

Response: Sorry, we are unable to find any spelling errors in those lines.  

Depression scale and life satifaction scale may be added as additionnal material. If not possible, a short specific description of items included in each scale could be provided in order to improve methodology description. 

Response:  We thanks the reviewer for their constructive suggestion. We have added a description of the depression scale (CESD-10) in the revised version of the manuscript (see lines 107-115). However, life satisfaction was assessed using a single item, which has already been explained in the manuscript.

Reviewer 2 Report

Article ID: ijerph-2071846

Title: Examining the association between sports participation and mental health of adolescents

General comments

Thank you for giving me the opportunity to review this paper.

This study aims: (i) to examine the relationship between sports participation and mental health of Bangladeshi adolescents, and (ii) to explore sports preferences and barriers to sports participation of both boys and girls.

I found the research question interesting and have no doubt that this study would be of interest to readers of this journal IJERPH, but qualitative analysis is not my expertise.

Some comments are included below.

_______________

Specific comments

ABSTRACT

Objective: to explore the association between sports participation and mental health in 320 adolescents (13-17 years; 59% boys) in Bangladesh. OR (i) to examine the relationship between sports participation and mental health of Bangladeshi adolescents, and (ii) to explore sports preferences and barriers to sports participation of both boys and girls. (?) Clarify in order to adjust with the last sentence in introduction.

INTRODUCTION (p.2-3)

Authors present: (P1) sports participation vs. health and wellbeing; (P2.) countries variability; P3) adolescent mental health problems, and (iv) Lack - limited research in adolescent mental health in Bangladesh. - OK

It seems relevant to adjust the objective. Suggestion: “The aims of the present study were: (i) to examine the relationship between sports participation and mental health of Bangladeshi adolescents, and (ii) to explore sports preferences and barriers to sports participation of both boys and girls.”

MATERIALS AND METHODS (p.3-4)

L82-86: present the ethical approval reference (number and date).

Quantitative examination - L123-131: Multiple linear regression modeling… quantitative variables? Present the used codification for the variables “Involved with team sport” and “Involved with non-team sports” in 2.1.5. Data analysis/MLR (lines:125-126). In my opinion, logistic regression (binary) should have been used.

Qualitative exploration: L148: “relevant literature review” // present some references.L159: translation validity ?? experts?

RESULTS (p.4-8)

Quantitative findings - L177-178: statistical results were presented in Table-1 (simplify the text). Also in L183-184). In accordance, present Table 1,.i.e.: “Results of MLR were presented in table 1.”

Qualitative findings - Sorry, but I’m not a specialist in qualitative analysis.

REFERENCES (p.11-12) #48...

Author Response

General comments

Thank you for giving me the opportunity to review this paper.

This study aims: (i) to examine the relationship between sports participation and mental health of Bangladeshi adolescents, and (ii) to explore sports preferences and barriers to sports participation of both boys and girls.

I found the research question interesting and have no doubt that this study would be of interest to readers of this journal IJERPH, but qualitative analysis is not my expertise.

Response: We thank the reviewer for their positive comments on our work.

Some comments are included below.

Specific comments

ABSTRACT

Objective: to explore the association between sports participation and mental health in 320 adolescents (13-17 years; 59% boys) in Bangladesh. OR (i) to examine the relationship between sports participation and mental health of Bangladeshi adolescents, and (ii) to explore sports preferences and barriers to sports participation of both boys and girls. (?) Clarify in order to adjust with the last sentence in introduction.

Response: Good suggestion – we have revised the aims accordingly (see lines 12-14).

INTRODUCTION (p.2-3)

Authors present: (P1) sports participation vs. health and wellbeing; (P2.) countries variability; P3) adolescent mental health problems, and (iv) Lack - limited research in adolescent mental health in Bangladesh. - OK

It seems relevant to adjust the objective. Suggestion: “The aims of the present study were: (i) to examine the relationship between sports participation and mental health of Bangladeshi adolescents, and (ii) to explore sports preferences and barriers to sports participation of both boys and girls.”

Response: Good suggestion – we have revised the relevant text accordingly (see lines 81-84).

MATERIALS AND METHODS (p.3-4)

L82-86: present the ethical approval reference (number and date).

Response: Good suggestion – we have added the approval numbers in the revised manuscript (see lines 87-89).

Quantitative examination - L123-131: Multiple linear regression modeling… quantitative variables? Present the used codification for the variables “Involved with team sport” and “Involved with non-team sports” in 2.1.5. Data analysis/MLR (lines:125-126). In my opinion, logistic regression (binary) should have been used.

Response: Sorry for the confusion caused. Both of our outcome/dependent variables (e.g., CESD-10 score and life satisfaction score) were continuous and as such multiple linear regression modelling was the best analytical approach to examine the relationship. We have tweaked the relevant text to clarify this. (see line 136)

We are aware that logistic regression could have been an ideal approach should we have had considered sports participation (yes/no, binary) as our dependent variable, but sports participation was our exposure or study factor.

Qualitative exploration: L148: “relevant literature review” // present some references.L159: translation validity ?? experts?

Response: We thank the reviewer for this useful suggestion.

Re L148, we have added a few references to support the development of focus group guide (see line 161).

Re L159, we have revised the relevant text with additional details to ensure the validity of translation from Bangla to English (see lines 172-74).     

RESULTS (p.4-8)

Quantitative findings - L177-178: statistical results were presented in Table-1 (simplify the text). Also in L183-184). In accordance, present Table 1,.i.e.: “Results of MLR were presented in table 1.”

Response: Good suggestion. We have revied the relevant results accordingly (see lines 190-91; 198-200).

Qualitative findings - Sorry, but I’m not a specialist in qualitative analysis.

Response: That should be fine as three of the four authors have worked in mixed method research, while the last author is an expert in qualitative research.

REFERENCES (p.11-12) #48...

Response: Many thanks!

Reviewer 3 Report

The study and the topic is very interesting. The theoretical framework is good and complete. The materials and methods section is very clear and well structured. The results are well defined. Perhaps it would be interesting to add some figures to visually represent the results.   Good conclusions.

Good work.

Author Response

The study and the topic is very interesting. The theoretical framework is good and complete. The materials and methods section is very clear and well structured. The results are well defined. Perhaps it would be interesting to add some figures to visually represent the results.   Good conclusions. Good work.

Response: We thank the reviewer for their appreciation of our work.